# The effect of SGLT2 inhibitor in patients with type 2 diabetes and atrial fibrillation

Yongin Cho[1], Sung-Hee Shin[2‡]*, Min-Ae Park[3], Young Ju Suh[4], Sojeong Park[3], Ji-Hun Jang[2], Dae-Young Kim[2], So Hun Kim[1‡]*

1 Department of Endocrinology and Metabolism, Inha University College of Medicine, Incheon, Republic of Korea, 2 Department of Cardiology, Inha University College of Medicine, Incheon, Republic of Korea, 3 Department of Data Science, Hanmi Pharm Co., Ltd, Republic of Korea, 4 Department of Biomedical Sciences, Inha University College of Medicine, Incheon, Korea

‡ SHS and SHK have contributed equally to this work and share corresponding authorship.
* sshin@inha.ac.kr (SHS); shoney@inha.ac.kr (SHK)

## Abstract

### Background

Sodium glucose cotransporter 2 (SGLT2) inhibitors improve clinical outcomes in several populations including type 2 diabetes (T2D), chronic renal insufficiency, and heart failure (HF). However, limited data exist on their effects on atrial fibrillation (AF).

### Methods

We conducted a retrospective cohort study using the National Health Insurance Service database. A total of 4,771 patients with T2D and AF who were newly prescribed SGLT2 inhibitors or DPP4 inhibitors were selected and matched in a 1:2 ratio by propensity score with 37 confounding variables. We assessed the effect of SGLT2 inhibitors on the composite outcome of either HF hospitalization or death.

### Results

Over a median follow-up of 31 months, patients on SGLT2 inhibitors were associated with a lower risk of hospitalizations for HF or mortality compared to those on DPP4 inhibitors (HR 0.61; 95% CI 0.44–0.85; P = 0.004). SGLT2 inhibitor use was also associated with a lower risk of mortality (HR 0.61; 95% CI 0.39–0.94; P = 0.025) and CV mortality (HR 0.43; 95% CI 0.21–0.86; P = 0.018), but not of MI (HR 1.22 [95% CI 0.72–2.09]; P = 0.461) or stroke (HR 1.00 [95% CI 0.75–1.33]; P = 0.980). The incidence of hospitalizations for HF, although statistically insignificant, tended to be lower in the SGLT2 inhibitor group (HR 0.63 [95% CI 0.39–1.02]; P = 0.062).

### Conclusion

In a nationwide cohort of patients with T2D and AF, SGLT2 inhibitor was associated with a lower risk of mortality, which may suggest that SGLT2 inhibitors may be considered as the first-line antidiabetic medication in patients with T2D and AF.

**Data Availability Statement:** The data supporting the findings of our study are provided by the Korea National Health Insurance Service (NHIS). However, there are ethical restrictions on sharing

the de-identified data set due to the presence of potentially identifiable or sensitive patient information. These restrictions are imposed by the NHIS in accordance with privacy and data protection regulations. Data are available upon request from the Korea National Health Insurance Service (NHIS) via their website (http://nhiss.nhis.or.kr) or telephone (+82-33-736-2431) for researchers who meet the criteria for access to confidential data.

**Funding:** This work was supported by a research grant from the Inha University.(Grant Number: Inha 70431) to S-H Shin. The funders had no role in study design, data collection and analysis, decision to publish, or preparation of the manuscript as mentioned in the manuscript.

**Competing interests:** The authors have declared that no competing interests exist.

## Introduction

Atrial fibrillation (AF), the most common sustained cardiac arrhythmia, is known to be associated with higher risks of stroke, heart failure (HF), and mortality [1]. A meta-analysis has demonstrated that patients with AF had a 4.62-fold higher risk of developing HF compared to those without AF [2]. Lack of atrial systole and irregular timing of diastolic can lead to elevated left atrial pressures and decreased stroke volume, which can facilitate the occurrence of HF [3]. Despite the contemporary therapeutic strategies, the burden of AF has remained constant and the risk of mortality in patients with AF has not improved substantially over the last decade [4].

Recently, sodium glucose co-transporter 2 (SGLT2) inhibitors have been shown to have clinical benefits in patients with type 2 diabetes at high cardiovascular (CV) risk, chronic renal insufficiency, or HF. This includes a reduction of the risk of hospitalization for HF, CV death, and improvement of renal outcomes [5]. While the mechanisms responsible for the beneficial effects of SGLT2 inhibitors are still under investigation, several mechanisms have been proposed such as reduced preload and afterload through osmotic diuresis and natriuresis, improving vascular function, improving cardiac energy metabolism, preventing inflammation, inhibiting the cardiac $Na+/H+$ exchange, and increasing erythytropoietin levels [6].

AF and diabetes often coexist, and when AF is present in patients with diabetes, it can lead to worse clinical outcomes. In a clinical trial with type 2 diabetic patients with established CV disease, those with AF at baseline had higher rates of adverse HF outcomes than those without AF. In this trial, SLGT2 inhibitors reduced HF-related and renal events irrespective of the presence of AF [7]. Also, several studies have reported that SGLT2 inhibitors reduced the burden of AF or atrial flutter [8].

However, data on the direct clinical effects of SGLT2 inhibitors in people with AF are still limited. In this study, we evaluate the clinical outcomes of SGLT2 inhibitors compared with DPP4 inhibitors in patients with type 2 diabetes and AF in a nationwide population-based cohort.

## Materials and methods

We analyzed the health records from the Korean Health Insurance Service database to estimate the effect of SGLT2 inhibitors on clinical outcomes. In this nationwide retrospective observational study, the database covers > 99% of the South Korean population. This contains all health records, including demographics, diagnoses coded with the International Classification of Diseases [ICD]-10, and drug prescriptions [9]. We used the data from January 1, 2008, to December 31, 2020 and did not have access to information that could identify individual participants during or after data collection. Data were accessed for research purpose in 17/10/2022. Our study protocol was reviewed and approved by the Institutional Review Board (IRB) of Inha University Hospital (2021-11-022), and informed consent was waived by the IRB.

### Study population

We included patients with both T2DM (ICD-10 code: E11-14 with at least one medication prescription history) and AF (ICD-10 code: I48) who were newly prescribed SGLT-2 inhibitors or DPP-4 inhibitors since 2014, which is when SGLT2 inhibitors were introduced into the South Korean Market. A new user was defined as a patient who had no history of medication of SGLT2 inhibitor or DPP4 inhibitor for at least 1 year and the first prescription date was designated the index date. We excluded the patients if they were diagnosed with chronic kidney disease stages 4 and 5 (ICD-10 code: N18.4–6), or any malignancy (ICD-10 code: C00-97).

## Clinical outcomes

The primary outcome for our analysis was a composite of the first occurrence of either hospitalization for HF (diagnosed as ICD-10 code I50 during the admission) or death. Secondary outcomes assessed included hospitalization for HF, all-cause mortality, CV mortality, stroke, myocardial infarction, a composite of hospitalization for HF or CV mortality, a composite of major cardiovascular events (MACE), defined as death, MI, or stroke, and hypoglycemia (Detailed definitions are described in S1 Table). The follow-up period was defined as the period from the index date until the occurrence of any of the clinical outcomes, discontinuation of the index drug, change to comparison drug or combined use, mortality, the latest follow-up date, or the end of the study period (December 31, 2020), whichever occurred first.

## Statistical analysis

Continuous values are presented as means (standard deviations; SD) and categorical values as numbers (percentages). To minimize differences in the baseline characteristics between the two groups, propensity score matching was performed. A total 37 matching variables included age, sex, duration of type 2 diabetes, index year, income, Charlson comorbidity index, prescribed drugs [1 year before the index date, particularly glucose-lowering medications and CVD medications], and comorbidities [1 year before the index date, particularly CVD diseases and microvascular diseases]. The nearest neighbor matching was used with a caliper (0.1). Propensity score matching was performed with a 1:2 ratio. The covariate balances between the two groups were calculated with standardized differences and absolute values < 0.1 (10%) of standardized differences were considered to be non-significant.

After propensity score matching, we performed survival analyses to estimate the effect of SGLT2 inhibitors on clinical outcomes. The Kaplan–Meier estimates were performed. The difference between the survival curves was assessed by the log-rank test. A marginal Cox proportional hazards regression models for a cluster in a matched pair were performed. We assessed the consistency of the drug effect on the primary outcome in subgroups. Subgroup analyses were performed to assess whether the effect of SGLT2 inhibitor on clinical outcomes varied across different baseline characteristics. We computed interaction p-values to detect effect modification. An interaction p-value < 0.05 would indicate a statistically significant interaction effect between the treatment and the subgroup variable, suggesting the presence of effect modification, but it is important to note that non-significant interaction p-values do not rule out the possibility of clinically relevant differences in subgroups [10]. In addition, to evaluate whether the benefit of the SGLT2 inhibitor varied with the follow-up period after the time of initiation, analyses were performed according to the time after initiation of the drug (30, 90, 180 days, 1, and 3 years after the index date). Two-sided p values < 0.05 were considered statistically significant. All analyses were performed with SAS (ver. 9.4; SAS Institute, Cary, NC, USA) and R software (ver. 4.0.3; R Development Core Team, Vienna, Austria).

## Result

Among 114,166 patients with type 2 diabetes and AF, a total of 23,467 new medication users were included in the cohort analysis (21,816 new users in the DPP4 inhibitor group and 1,616 new users in the SGLT2 inhibitor group). After propensity score matching, 4,771 patients were finally included (Fig 1). Table 1 showed baseline characteristics before and after propensity score matching. The standardized differences in all variables were<0.1 (10%) after propensity score matching, showing the characteristics were well-balanced between the two groups.

Over the median follow-up of 31 months (interquartile range, 8 to 75 months), 181 patients (3.7%) died, 70 patients (1.5%) as a result of a CV cause, 121 patients (2.5%) were hospitalized

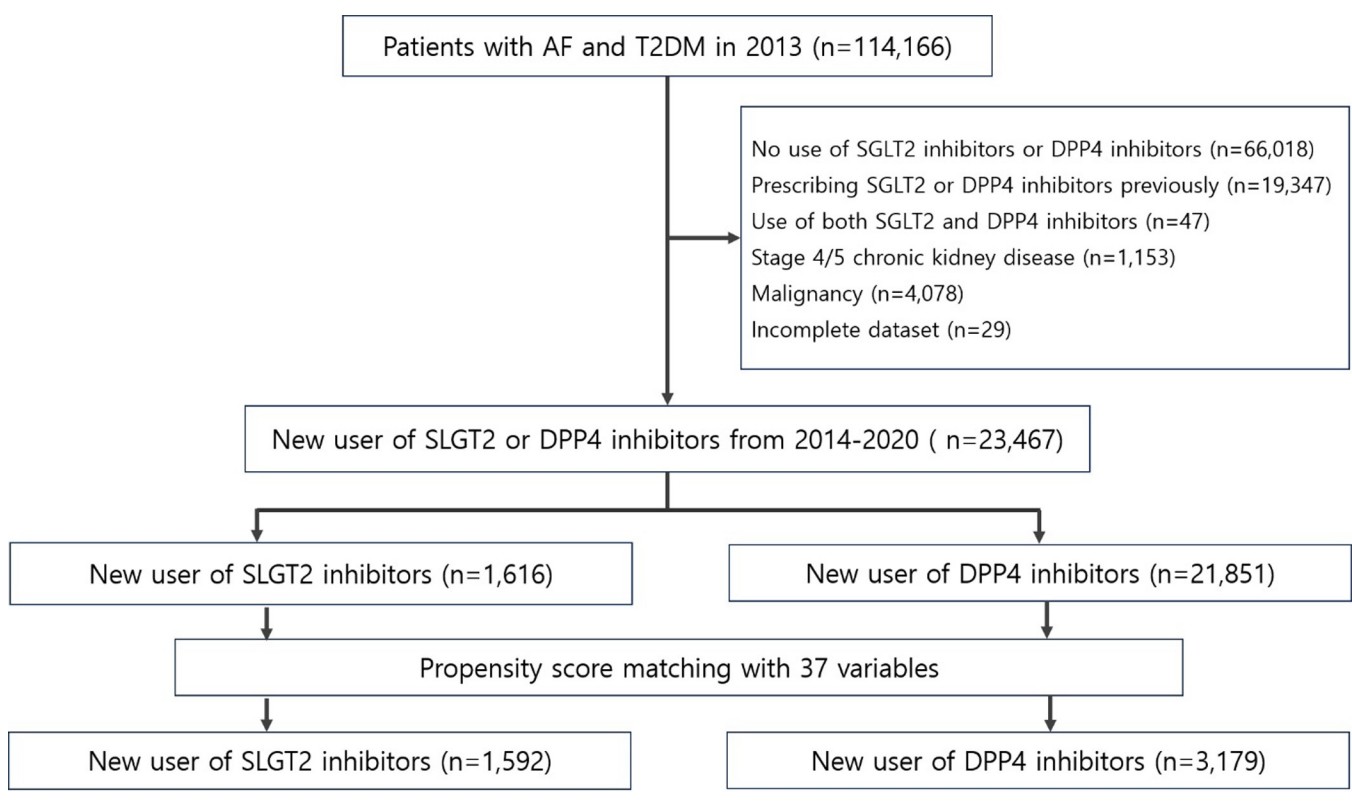

**Fig 1. Flow chart of patient selection.**

for HF, and 283 patients (5.9%) were hospitalized for HF or died. 95 patients (2.0%) experienced MI, 293 patients (6.1%) had stroke, and 496 patients (10.4%) experienced MACE. In both the Kaplan-Meier and Cox proportional hazards models, patients treated with SGLT2 inhibitor showed a lower risk of HF hospitalization or mortality than those treated with DPP4 inhibitor (HR 0.61 [95% CI 0.44–0.85], P = 0.004, Table 2 and Fig 2). Additionally, the use of SGLT2 inhibitor was associated with lower risk of mortality (HR 0.61; 95% CI 0.39–0.94; P = 0.025), including CV mortality (HR 0.43 [95% CI 0.21–0.86], P = 0.018), but not of MI (HR 1.22 [95% CI 0.72–2.09]; P = 0.461), stroke (HR 1.0 [95% CI 0.75–1.33], P = 0.980), MACE (HR 0.98 [95% CI 0.78–1.23], P = 0.831) or hypoglycemia (HR 0.83 [95% CI 0.51–1.35], P = 0.446) compared to DPP4 inhibitor. Although the incidence of hospitalizations for HF tended to be lower in the SGLT2 inhibitor group according to Kaplan-Meier analysis (Fig 2), it did not reach a statistically significant level in the Cox regression model (HR 0.63 [95% CI 0.39–1.02]; P = 0.062, Table 2).

The effect of the SGLT2 inhibitor on the primary outcome was consistent across all subgroups, indicating that SGLT2 inhibitors were associated with a lower risk of the composite of HF hospitalization or death compared to DPP4 inhibitors, regardless of age, gender, comorbidities, prior history of atherosclerotic CV disease or HF, and the use of beta blocker or renin-angiotension system blocker (all p for interaction>0.05, Fig 3). This suggests that the impact of SGLT2 inhibitor relative to the DPP-4 inhibitor was consistent across the baseline characteristics evaluated. However, this should not be interpreted as definitive evidence of uniform effects across all subgroups. Further studies with more power or focusing on specific subgroups may provide additional insights. Patients with SGLT2 inhibitor demonstrated a

**Table 1. Baseline characteristics of patients before and after propensity score matching.**

| | Entire population (n = 23,467) | | | | Propensity score matched population (n = 4,771) | | | |
|---|---|---|---|---|---|---|---|---|
| | SGLT2i (n = 1,616) | DPP4i (n = 21,851) | P value | SMD | SGLT2i (n = 1,592) | DPP4i (n = 3,179) | P value | SMD |
| Mean (SD) age (years) | 67.9 (10.7) | 71.8 (10.1) | <0.001 | -0.380 | 67.9 (10.7) | 68.0 (10.7) | 0.702 | -0.012 |
| Female | 636 (39%) | 9927 (45%) | <0.001 | 0.128 | 623 (39%) | 1261 (40%) | 0.722 | 0.011 |
| Diabetes duration | | | <0.001 | 0.745 | | | 0.938 | -0.002 |
| ≥6 years | 283 (18%) | 5445 (25%) | | | 278 (17%) | 558 (18%) | | |
| <6 years | 1333 (82%) | 16406 (75%) | | | 1314 (83%) | 2621 (82%) | | |
| Index year | | | <0.001 | 0.736 | | | 0.997 | 0.003 |
| 2014–2015 | 289 (78%) | 10760 (49%) | | | 285 (18%) | 567 (18%) | | |
| 2016–2017 | 550 (34%) | 6456 (30%) | | | 540 (34%) | 1089 (34%) | | |
| 2018–2019 | 528 (33%) | 3476 (16%) | | | 523 (33%) | 1039 (33% | | |
| 2020 | 249 (15%) | 1159 (5%) | | | 244 (15%) | 484 (15%) | | |
| Income | | | 0.087 | -0.003 | | | 0.884 | -0.006 |
| Very low | 357 (22%) | 4866 (22%) | | | 357 (22%) | 701 (22%) | | |
| Low | 243 (15%) | 2887 (13%) | | | 243 (15%) | 475 (15%) | | |
| High | 315 (19%) | 4063 (19%) | | | 315 (20%) | 660 (21%) | | |
| Very high | 678 (42%) | 9717 (44%) | | | 677 (43%) | 1343 (42%) | | |
| Missing | 23 (1%) | 318 (1%) | | | | | | |
| Charlson comorbidity index | | | <0.001 | -0.330 | | | 0.922 | 0.012 |
| 0 | 1209 (75%) | 13131 (60%) | | | 1187 (75%) | 2387 (75%) | | |
| 1 | 131 (8%) | 2335 (11%) | | | 131 (8%) | 258 (8%) | | |
| ≥2 | 276 (17%) | 6385 (29%) | | | 274 (17%) | 534 (17%) | | |
| Underlying disease | | | | | | | | |
| Hypertension | 1435 (89%) | 18240 (83%) | <0.001 | -0.155 | 1414 (89%) | 2381 (89%) | 0.808 | 0.007 |
| Coronary artery disease | 676 (42%) | 8565 (39%) | 0.037 | 0.056 | 668 (42%) | 1337 (42%) | 0.949 | -0.024 |
| Heart Failure | 655 (41%) | 8124 (37%) | 0.007 | -0.065 | 641 (40%) | 1298 (41%) | 0.707 | 0.012 |
| Stroke | 313 (19%) | 5656 (26%) | <0.001 | 0.154 | 310 (19%) | 614 (19%) | 0.896 | -0.004 |
| Peripheral artery disease | 356 (22%) | 5781 (26%) | <0.001 | 0.108 | 345 (22%) | 751 (24%) | 0.172 | 0.041 |
| Dyslipidemia | 1359 (84%) | 18427 (84%) | 0.803 | 0.009 | 1337 (84%) | 2664 (84%) | 0.872 | -0.005 |
| Chronic kidney disease | 46 (2.9%) | 1415 (6.5%) | <0.001 | 0.113 | 46 (3%) | 112 (4%) | 0.249 | 0.030 |
| Microvascular disease | | | | | | | | |
| Diabetic neuropathy | 316 (20%) | 5305 (24%) | <0.001 | 0.113 | 313 (20%) | 639 (20%) | 0.720 | 0.011 |
| Diabetic retinopathy | 326 (20%) | 4888 (22%) | 0.040 | 0.053 | 322 (20%) | 652 (21%0 | 0.819 | 0.007 |
| Diabetic nephropathy | 198 (12%) | 3066 (14%) | 0.046 | 0.052 | 196 (12%) | 406 (13%) | 0.652 | 0.014 |
| Glucose-lowering therapies | | | | | | | | |
| Insulin | 261 (16%) | 5094 (23%) | <0.001 | 0.178 | 260 (16%) | 493 (16%) | 0.462 | -0.020 |
| Metformin | 1113 (69%) | 15329 (70%) | 0.279 | 0.031 | 1094 (69%) | 2188 (69%) | 0.939 | 0.002 |
| Sulfonylurea | 627 (39%) | 10613 (49%) | <0.001 | 0.197 | 619 (39%) | 1206 (38%) | 0.526 | -0.019 |
| Thiazolidinedione | 124 (8%) | 1428 (7%) | 0.076 | -0.047 | 123 (8%) | 493 (8%) | 0.588 | 0.018 |
| GLP-1 receptor agonist | 4 (0.3%) | 12 (0.05%) | 0.004 | -0.050 | 4 (0.3%) | 8 (0.3%) | 0.998 | 0 |
| Alpha glucosidase inhibitor | 101 (6%) | 1982 (9%) | <0.001 | 0.108 | 98 (6%) | 199 (6%) | 0.889 | 0.004 |
| Cardiovascular therapies | | | | | | | | |
| Antithrombotic agents | 1409 (87%) | 18863 (86%) | 0.327 | -0.028 | 1390 (87%) | 2790 (88%) | 0.655 | 0.013 |
| ACE inhibitor or ARB | 1154 (71%) | 14254 (65%) | <0.001 | -0.133 | 1138 (71%) | 2271 (71%) | 0.974 | -0.001 |
| Beta blocker | 919 (57%) | 11294 (52%) | <0.001 | -0.106 | 906 (57%) | 1800 (57%) | 0.850 | 0.013 |
| Calcium channel blocker | 771 (48%) | 10520 (48%) | 0.736 | 0.011 | 757 (48%) | 1532 (48%) | 0.676 | 0.013 |
| Stain | 1178 (73%) | 13921 (64%) | <0.001 | -0.028 | 1161 (73%) | 2323 (73%) | 0.915 | 0.003 |

(*Continued*)

**Table 1.** (Continued)

| | Entire population (n = 23,467) | | | | Propensity score matched population (n = 4,771) | | | |
|---|---|---|---|---|---|---|---|---|
| | SGLT2i (n = 1,616) | DPP4i (n = 21,851) | P value | SMD | SGLT2i (n = 1,592) | DPP4i (n = 3,179) | P value | SMD |
| Diuretics | 879 (54%) | 12358 (56%) | 0.091 | 0.038 | 871 (55%) | 1689 (53%) | 0.302 | 0.005 |
| Antiarrhythmic drug | 578 (36%) | 8627 (39%) | 0.003 | 0.081 | 565 (35%) | 1156 (36%) | 0.512 | 0.020 |

Values are numbers (percentages) unless stated otherwise.

ACE; angiotensin converting enzyme, ARB; angiotensin receptor blocker, DPP4i; dipeptidyl peptidase 4 inhibitors, GLP; glucagon-like peptide, SD; standard deviation, SGLT2i; sodium glucose cotransporter-2 inhibitors, SMD; standardized mean difference

significantly lower risk of primary outcome from 90 days after initiating the drug to the third year, as compared with those with DPP4 inhibitor (P<0.05, Table 3).

## Discussion

In this nationwide population-based cohort study, we found that SGLT2 inhibitors were associated with a lower risk of the composite outcome of hospitalization for HF or mortality compared to DPP4 inhibitors in patients with both type 2 diabetes and AF. Use of SGLT2 inhibitors was also associated with a lower risk of mortality including CV mortality, but not with a lower risk of stroke, MI, MACE, or hypoglycemia. The incidence of hospitalizations for HF tended to be lower in the SGLT2 inhibitor group but not statistically significant in the Cox regression model.

In our study, we selected DPP4 inhibitors as an active comparator like other prior studies, because they are a relatively new and widely used oral antihyperglycemic agent [9, 11]. Like SGLT2 inhibitors, DPP4 inhibitors are commonly used as a second-line therapy after metformin in Korea [12]. Both DPP4 inhibitors and SGLT2 inhibitors are prescribed under similar clinical circumstances, specifically for patients who do not achieve sufficient glycemic control

**Table 2. Event rates and hazard ratios for clinical outcomes.**

| Variables | SGLT2 inhibitor | | DPP4 inhibitor | | | |
|---|---|---|---|---|---|---|
| | No of events | Incidence rate /1000 person-years(95% CI) | No of events | Incidence rate /1000 person-years(95% CI) | Hazard ratio (95% CI) | P-value |
| Outcomes of interest: | | | | | | |
| HHF+Mortality | 60 | 27.8 (21.6–35.8) | 223 | 43.1 (37.8–49.2) | 0.61 (0.44–0.85) | 0.004 |
| HHF+CV mortality | 37 | 17.1 (12.4–23.6) | 143 | 27.7 (23.5–32.6) | 0.55 (0.36–0.82) | 0.004 |
| HHF | 27 | 12.5 (8.6–18.2) | 94 | 18.2 (14.9–22.3) | 0.63 (0.39–1.02) | 0.062 |
| All-cause mortality | 37 | 17.0 (12.3–23.4) | 144 | 27.4 (23.2–32.2) | 0.61 (0.39–0.94) | 0.025 |
| CV mortality | 13 | 6.0 (3.5–10.3) | 57 | 10.8 (8.4–14.0) | 0.43 (0.21–0.86) | 0.018 |
| Stroke | 89 | 42.0 (34.1–51.7) | 204 | 40.2 (35.1–46.1) | 1.00 (0.75–1.33) | 0.980 |
| MI | 31 | 14.4 (10.1–20.4) | 64 | 12.3 (9.6–15.7) | 1.22 (0.72–2.09) | 0.461 |
| MACE | 146 | 69.5 (59.1–81.7) | 350 | 69.5 (62.5–77.1) | 0.98 (0.78–1.23) | 0.831 |
| Hypoglycemia | 28 | 13.1 (9.0–18.9) | 86 | 16.7 (13.5–20.6) | 0.83 (0.51–1.35) | 0.446 |

CI; confidence interval, CV; Cardiovascular, DPP4i; dipeptidyl peptidase 4 inhibitors, HHF; hospitalization for heart failure, MACE; major cardiovascular events, MI; Myocardial infarction, SGLT2i; sodium glucose cotransporter-2 inhibitors

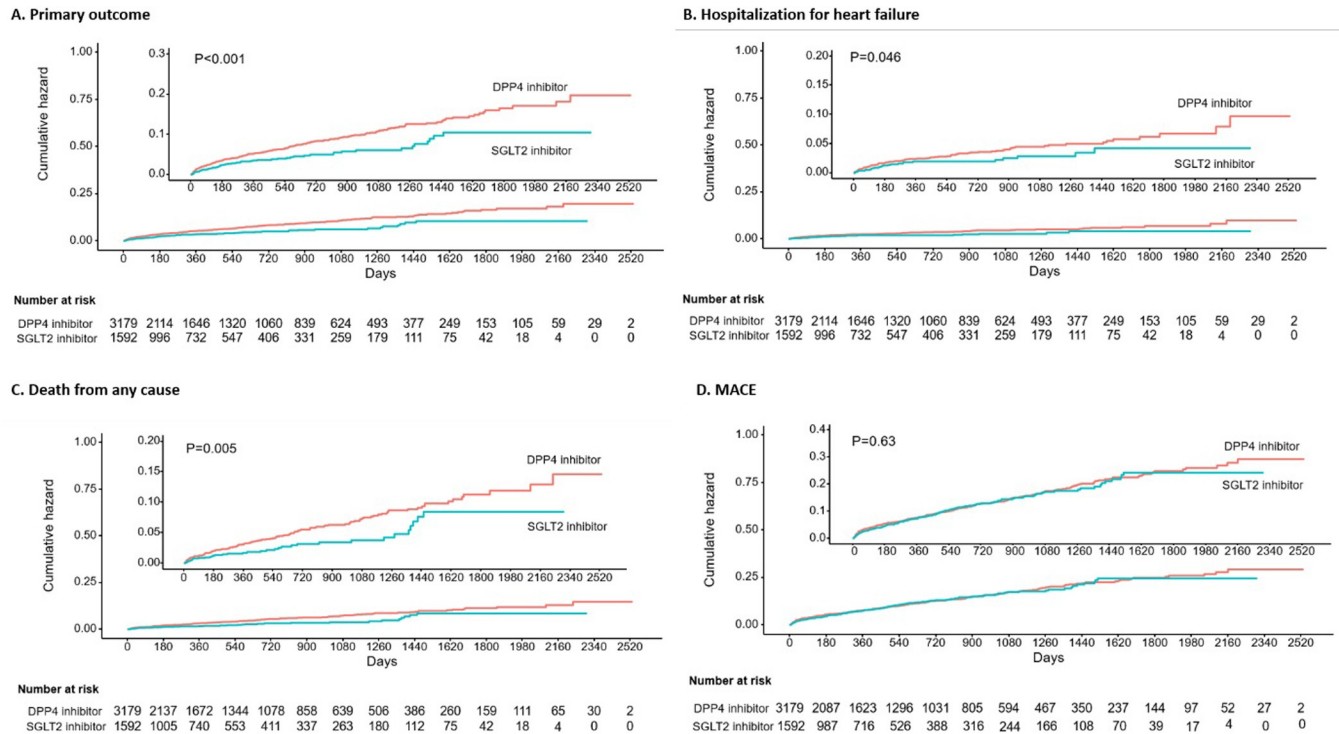

**Fig 2. Time to event curves for clinical outcomes.** Kaplan–Meier plots of each clinical outcome. A. Primary outcome of hospitalization for heart failure or death from any cause. B. Hospitalization for heart failure. C. Death from any cause. D. A composite of major cardiovascular events. MACE; A composite of major cardiovascular events.

with metformin alone. This similarity would help minimize the potential for confounding in our evaluation of CV outcomes. In addition, DPP4 inhibitors have shown a neutral effect in most CV outcome trials [13, 14], and they have a moderate effect in reducing HbA1c with a low risk of hypoglycemia similar to SGLT2 inhibitors [15]. By choosing DPP-4 inhibitors, we aimed to compare the efficacy and safety of SGLT2 inhibitors against a drug with a similar clinical profile, thereby providing meaningful insights into their relative effects.

AF and diabetes are often comorbid conditions, which can be linked through oxidative stress and inflammatory mechanisms. Diabetes can promote the development and maintenance of AF by exacerbating atrial electrical and structural remodeling [16]. The early stages of diabetes-induced myocardial changes are characterized by increased fibrosis and stiffness, which is reflected by decreased early diastolic filling, increased atrial filling and dilatation, and elevated left ventricular end-diastolic pressure [17]. Also, in patients with AF, it has been reported that those with diabetes have a worse quality of life and more clinical events including all-cause or CV mortality and hospitalizations, compared to patients without diabetes [18, 19]. Thus, a holistic approach would be important in managing patients with concurrent AF and diabetes.

Clinical trials have shown the beneficial effects of SGLT2 inhibitors in patients with type 2 diabetes at high CV risk or established atherosclerotic CV diseases, HF, or chronic kidney disease. A meta-analysis indicated that SGLT inhibitors resulted in a lower risk in the composite of hospitalization for HF or CV death by 30% in patients with a history of AF, which was similar to the effect estimate for patients without AF [20]. However, landmark clinical trials included diabetic patients at high risk or specific population, and it may not be generalizable to all the patients with AF in the community. In our study, we demonstrated consistent clinical

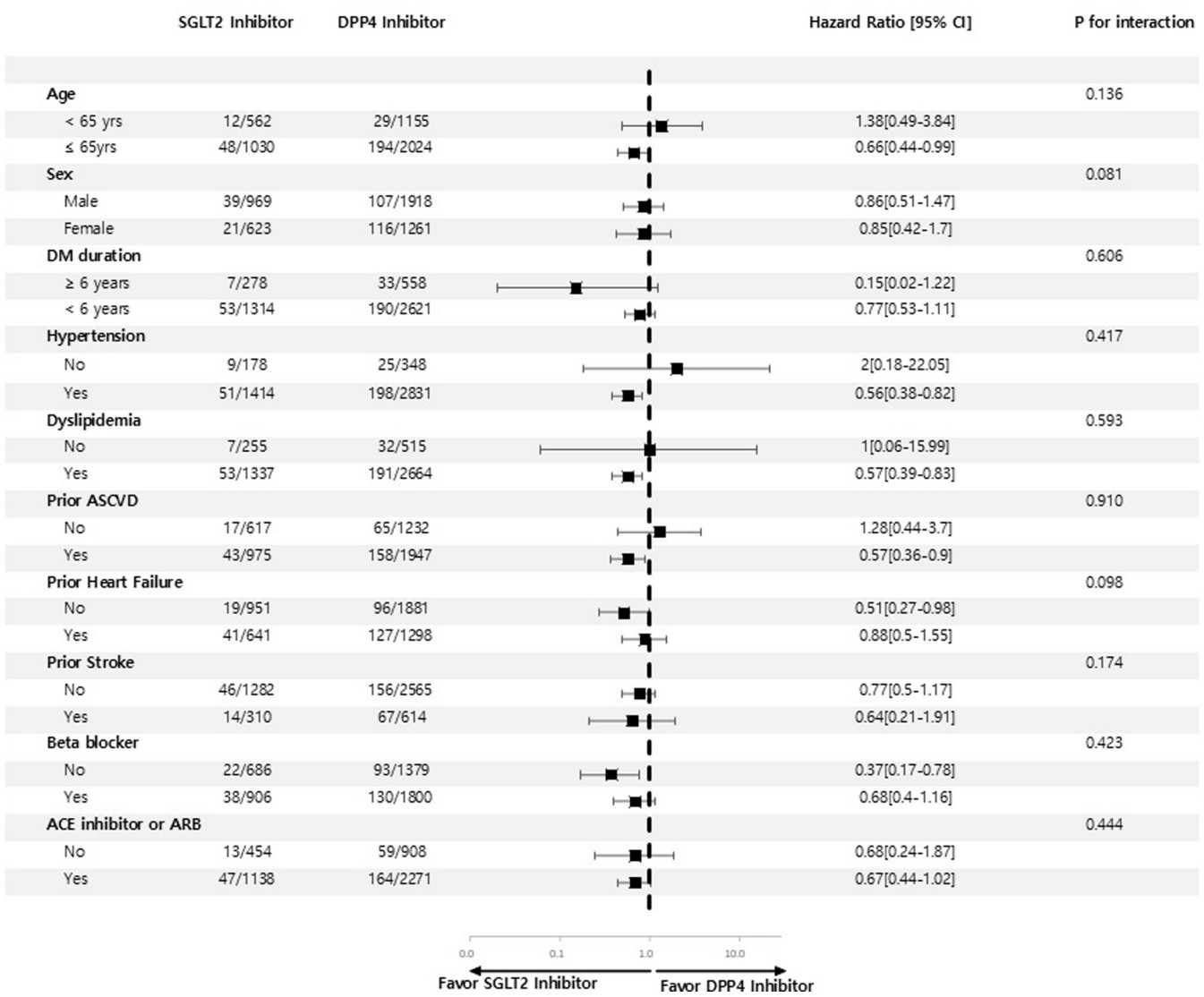

**Fig 3. Primary outcomes according to subgroups.** Subgroup analysis of forest plot of hazard ratio for SGLT2 inhibitors versus DPP4 inhibitors. Subgroup analysis showed consistent results for a lower risk of the primary outcome of hospitalization for heart failure or death from any cause for SGLT2 inhibitors vs. DPP4 inhibitors (all P interaction > 0.05). HR; hazard ratio, ACE; angiotension converting enzyme, ARB; angiotension receptor blocker, ASCVD; atherosclerotic cardiovascular disease.

**Table 3. The hazards of hospitalization for heart failure or mortality according to follow-up period.**

| Time after drug initiation | Event, no/ Number at risk | | Hazard ratio [95% CI] | P value |
|---|---|---|---|---|
| | SGLT2 inhibitor | DPP4 inhibitor | | |
| 30 days | 7/204 | 28/398 | 0.46 [0.20–1.06] | 0.070 |
| 90 days | 16/410 | 47/725 | 0.56 [0.32–0.98] | 0.044 |
| 180 days | 22/592 | 73/1048 | 0.52 [0.32–0.84] | 0.008 |
| 1 year | 33/859 | 105/1523 | 0.56 [0.38–0.83] | 0.004 |
| 2 years | 44/1188 | 155/2112 | 0.51 [0.36–0.71] | <0.001 |
| 3 years | 50/1338 | 182/2557 | 0.59 [0.43–0.80] | <0.001 |

CI, confidence interval

benefits of SGLT2 inhibitors compared to DPP4 inhibitors in patients with AF and type 2 diabetes using a nationwide cohort with long-term follow-up. And this clinical benefit became apparent from 90 days after initiating the drug. Interestingly, the benefit of the SGLT2 inhibitor was more pronounced for mortality than for HF hospitalization. While all-cause mortality was included as a component of the primary composite endpoint rather than CV mortality in our analysis because there is a potential for misclassification of cause of death due to the characteristics of this nationwide cohort, CV mortality also showed a significantly lower rate in the SGLT2 inhibitor group.

Multiple mechanisms have been proposed for the beneficial effects of SGLT2 inhibitors. SGLT-2 inhibition causes glycosuria, as well as natriuresis, osmotic diuresis, and plasma volume constriction [21]. This results in beneficial effects on parameters such as glucose concentration, body weight, blood pressure, and albuminuria, which are also associated with reduced risk of CV death and HF [22]. Recently, other cardioprotective mechanisms have also been discussed. Mechanisms such as prevention of cardiac inflammation and oxidative stress, apoptosis, dysfunction of ionic homeostasis, and mitochondrial dysfunction have been proposed [23, 24]. These effects could potentially extend to patients with AF. In addition, the reduction of AF burden by SGLT2 inhibitor might be attributable to more favorable clinical outcomes. Previous studies have shown that the use of SGLT2 inhibitors was associated with a reduced risk of incident AF [7, 8, 25–28]. Osmotic diuretic and natriuresis will cause a reduction in blood pressure and body weight, which subsequently may retard arterial remodeling. Animal study showed SGLT2 inhibitors can ameliorate arterial structural and electrical remodeling through the improvement of mitochondrial function and mitochondrial biogenesis [29].

Several limitations of our study should be noted. First, the national cohort database used in our analysis provides drug information only by class to protect personal information. Therefore, we were unable to assess the effects according to the detailed types of each drug used. Also, detailed patient information, type of AF, and laboratory results were not available, which did not allow us to conduct analyses related to these aspects. Thus, we could not perform detailed analysis to explore the mechanisms through which SGLT2 inhibitors exert their effects, such as potential associations with blood pressure, lipid profiles, or diuretic effects. Second, concerns about the risk of HF for some of the DPP4 inhibitors have not been completely resolved [30, 31]. Therefore, the use of DPP4 inhibitors, an active comparator, may have been associated with an increased risk of HF. However, there is no evidence that DPP4 inhibitors are associated with an increased risk of mortality so far. Third, clinical outcomes were ascertained from the national health insurance service database and national statistical database, but not adjudicated from each patient's medical records or laboratory tests. Despite prior study having shown favorable reliability of primary diagnostic codes of major clinical outcomes in this database, there remains the possibility of misclassification of clinical events. Fourth, we assessed the effects of SGLT inhibitors in diabetic patients with AF. Thus, whether SGLT2 inhibitors improve clinical outcomes in non-diabetic patients with AF is uncertain. Further research is warranted to explore the effects of SGLT2 inhibitors in patients with AF, irrespective of their diabetes status. Finally, baseline characteristics were different between the SGLT2 inhibitor group and the DPP4 inhibitor group. Although efforts are made to balance clinical characteristics by propensity score matching with 37 variables, residual confounders can still remain.

## Conclusion

In patients with both AF and type 2 diabetes, SGLT2 inhibitors were associated with a lower risk of either hospitalization for HF or mortality compared to DPP4 inhibitors, which may

suggest that SGLT2 inhibitors may be considered as the first-line antidiabetic medication in patients with AF and diabetes.

## Supporting information

**S1 Table. The definitions and codes used for each outcome.**
(DOCX)

## Author Contributions

**Conceptualization:** Yongin Cho, Sung-Hee Shin, Ji-Hun Jang, Dae-Young Kim, So Hun Kim.

**Data curation:** Yongin Cho, Sung-Hee Shin, Young Ju Suh, Sojeong Park.

**Formal analysis:** Min-Ae Park, Young Ju Suh, Sojeong Park.

**Funding acquisition:** Sung-Hee Shin.

**Investigation:** Yongin Cho, Sung-Hee Shin, Min-Ae Park, So Hun Kim.

**Methodology:** Yongin Cho, Sung-Hee Shin, Young Ju Suh.

**Supervision:** Sung-Hee Shin, Young Ju Suh, So Hun Kim.

**Writing – original draft:** Yongin Cho, Sung-Hee Shin.

**Writing – review & editing:** Sung-Hee Shin, Ji-Hun Jang, Dae-Young Kim, So Hun Kim.

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
