## [Decision Letter · Decision Letter 0]

24 Jul 2024

PONE-D-24-09659The effect of SGLT2 inhibitor in patients with type 2 diabetes and atrial fibrillationPLOS ONE

Dear Dr. Shin,

Thank you for submitting your manuscript to PLOS ONE. After careful consideration, we feel that it has merit but does not fully meet PLOS ONE’s publication criteria as it currently stands. Therefore, we invite you to submit a revised version of the manuscript that addresses the points raised during the review process.

We look forward to receiving your revised manuscript.

Kind regards,

Xinlin Zhang

Academic Editor

PLOS ONE

“This work was supported by a research grant from the Inha University.(Grant Number: Inha 70431) to S-H Shin.”

Reviewers' comments:

Reviewer's Responses to Questions

**Comments to the Author**

1. Is the manuscript technically sound, and do the data support the conclusions?

Reviewer #1: Yes

Reviewer #2: Yes

2. Has the statistical analysis been performed appropriately and rigorously? 

Reviewer #1: Yes

Reviewer #2: Yes

3. Have the authors made all data underlying the findings in their manuscript fully available?

Reviewer #1: Yes

Reviewer #2: Yes

4. Is the manuscript presented in an intelligible fashion and written in standard English?

Reviewer #1: Yes

Reviewer #2: Yes

5. Review Comments to the Author

Reviewer #1: 1. In the results section, the authors stated “The effect of the SGLT2 inhibitor on the primary outcome was consistent across all subgroups (Figure 3)”. This sentence does not reflect whether the results in each subgroup were significant or not. I recommend adding a much clearer statement about this result.

2. In the discussion section, the authors stated “we selected DPP4 inhibitors as an active comparator like other prior studies because it is a relatively new and widely used oral antihyperglycemic agent”. Authors should improve justification on why DPP4 inhibitors are an adequate comparator to evaluate effect differences from SGLT-2 inhibitors. Also, authors should give reasons on why they do not make comparisons with other pleiotropic drugs such as Metformin or GLP-1 analogs. I suggest, if there is available data from the cohort, add analyses including all other anti-diabetic medications as a “usual therapy” control group in the supplement.

3. In the discussion section, you stated “DPP4 inhibitors are the most commonly used drug in Korea as a second-line therapy after metformin”. If DPP4 inhibitors are considered a second line medication in your study population, diabetic patients prescribed with this medication may be less controlled overall, which may be a source of confound if you are evaluating the effect in CV outcomes.

4. In the discussion section, the authors used the following statement “Several studies have shown that the use of SGLT2 inhibitors was associated with a reduced risk of incident AF”. I recommend rewriting the sentence avoiding the word “several” as you only cite 2 studies. Reference 24 is a review of individual studies, I suggest citing all of them instead of just citing the review.

5. Finally, in the discussion section you make an interesting acknowledgement about the limitation of your study to assess the effect in SGLT-2 inhibitors in non-diabetic patients with AF. I recommend adding a suggestion to further research in this population.

Reviewer #2: This research presents a retrospective cohort study investigating the efficacy of SGLT2i in patients with type 2 diabetes (T2D) complicated by atrial fibrillation (AF). Leveraging the Korean National Health Insurance Service database, the study assesses the impact of SGLT2i on the composite endpoint of hospitalization for heart failure (HF) or death. The merits of this research lie in: 1) Demonstrating that SGLT2i significantly reduce the risk of HF hospitalization or death among T2D patients with AF, providing novel evidence for the application of SGLT2i in this patient population. 2) Conducting a large-scale retrospective study utilizing the national health insurance database, enhancing the reliability and representativeness of the findings. 3) Implementing propensity score matching for 37 confounding variables, effectively mitigating their influence and ensuring the rigor of the study design.

However, I have the following questions regarding this research:

1. The lack of detailed subgroup analyses by different types of AF (e.g., paroxysmal, persistent, or permanent) or by specific SGLT2i limits the generalizability of the results.

2. Given that SGLT2i can improve blood pressure, lipid profiles, and exhibit diuretic effects, all of which may contribute to better AF outcomes, could mediation analysis be employed to explore the role of these improvements in mediating the observed effects?

3. Subgroup analyses based on patient baseline characteristics are crucial to gain a fuller understanding of the efficacy variations of SGLT2i across different patient groups.

4. The situation of combination therapy should be carefully elucidated. GLP-1RA use could impact cardiac outcomes. While SGLT2i patients likely have GLP-1RA co-treatment, DPP4i patients should not. Yet, Table 1 shows 8% GLP-1RA co-use in both, raising doubts about a statistical error.

6. PLOS authors have the option to publish the peer review history of their article (what does this mean?). If published, this will include your full peer review and any attached files.

Reviewer #1: **Yes: **Marcelo Reategui-Diaz

Reviewer #2: **Yes: **Xiaowen Zhang

---

## [Author Response · Author response to Decision Letter 0]

2 Sep 2024

Reviewer #1: 

1. In the results section, the authors stated “The effect of the SGLT2 inhibitor on the primary outcome was consistent across all subgroups (Figure 3)”. This sentence does not reflect whether the results in each subgroup were significant or not. I recommend adding a much clearer statement about this result. 

First, we sincerely appreciate all your constructive and valuable comments. 

In our subgroup analysis, SGLT2 inhibitors showed a lower risk in the composite of HF hospitalization or death than DPP2 inhibitors regardless of age, gender, comorbidities, prior history of ASCVD or heart failure, and use of beta blocker or RAS blocker. The p value for interaction for all these subgroups were greater than 0.05. We included this in the revised manuscript for clearer understanding as the reviewer suggested. 

Revision in the manuscript on page 8: 

The effect of the SGLT2 inhibitor on the primary outcome was consistent across all subgroups, indicating that SGLT2 inhibitors were associated with a lower risk of the composite of HF hospitalization or death compared to DPP2 inhibitors, regardless of age, gender, comorbidities, prior history of atherosclerotic CV disease or HF, and the use of beta blocker or renin-angeiotensin system blocker (all p for interaction>0.05, Figure 3).

2. In the discussion section, the authors stated “we selected DPP4 inhibitors as an active comparator like other prior studies because it is a relatively new and widely used oral antihyperglycemic agent”. Authors should improve justification on why DPP4 inhibitors are an adequate comparator to evaluate effect differences from SGLT-2 inhibitors. Also, authors should give reasons on why they do not make comparisons with other pleiotropic drugs such as Metformin or GLP-1 analogs. I suggest, if there is available data from the cohort, add analyses including all other anti-diabetic medications as a “usual therapy” control group in the supplement.

We thank to the reviewer for the insightful comments. Unfortunately, since we received data exclusively on users of SGLT2 inhibitors and DPP4 inhibitors from the National Health Insurance Service, it is difficult to perform a comparative analysis between SGLT2 inhibitors and all other anti-diabetic medications with the current data, as the reviewer suggested.

We chose DPP4 inhibitors as an active comparator in our study, similar to other prior studies, because it is a relatively new and widely used oral antihyperglycemic agent. Indeed, DPP4 inhibitors are the most commonly used second line drug after metformin in Korea. In addition, they have shown a neutral effect in most CV outcome trials and have a moderate effect in reducing HbA1c with a low risk of hypoglycemia, similar to SGLT2 inhibitors.

While most medical costs are covered by the National Health Insurance Service in South Korea, the National Health Insurance system in Korea generally recommends metformin as the first-line treatment for diabetes. Other antidiabetic medications, including SGLT2 inhibitors, are typically used as second-line treatments after metformin. Therefore, we considered that comparing SGLT2 inhibitors with metformin may be inappropriate due to significant differences in their recommended indications and usage patterns, which would not allow for proper patient matching. Moreover, other medications such as insulin, thiazolidinediones, GLP-1 receptor agonists, and alpha-glucosidase inhibitors are used less frequently in patients with diabetes and atrial fibrillation (less than 10% usage) in Korea. This limited use poses challenges for robust statistical analysis. Although Sulfonylureas are used more frequently, they have a different hypoglycemia risk profile compared to SGLT2 inhibitors and are subject to different insurance reimbursement criteria, complicating direct comparisons. We added this in the revised manuscript as the following. 

Revision in the manuscript on page 9-10: 

In our study, we selected DPP4 inhibitors as an active comparator like other prior studies, because they are relatively new and widely used oral antihyperglycemic agents.9 10 Like SGLT2 inhibitors, DPP4 inhibitors are commonly used as second-line therapies after metformin in Korea. 11 Both DPP-4 inhibitors and SGLT2 inhibitors are prescribed under similar clinical circumstances, specifically for patients who do not achieve sufficient glycemic control with metformin alone. This similarity would help reduce the potential for confounding in our evaluation of CV outcomes. In addition, DPP4 inhibitors have shown a neutral effect in most CV outcome trials,12 13 and they have a moderate effect in reducing HbA1c with a low risk of hypoglycemia, similar to SGLT2 inhibitors.14 By choosing DPP-4 inhibitors, we aimed to compare the efficacy and safety of SGLT2 inhibitors against a drug with a similar clinical profile, thereby providing meaningful insights into their relative effects.

3. In the discussion section, you stated “DPP4 inhibitors are the most commonly used drug in Korea as a second-line therapy after metformin”. If DPP4 inhibitors are considered a second line medication in your study population, diabetic patients prescribed with this medication may be less controlled overall, which may be a source of confound if you are evaluating the effect in CV outcomes.

We apologize for any confusion caused by our previous explanation. As noted, in Korea, DPP-4 inhibitors are used as a second-line therapy following metformin, and SGLT2 inhibitors are similarly used as second-line treatments when metformin alone does not achieve adequate glycemic control. In clinical practice, both medications are considered for patients whose glycemic levels remain inadequately controlled with metformin alone. Therefore, within the context of our study, the clinical scenarios in which DPP-4 inhibitors and SGLT2 inhibitors are prescribed are comparable.

It is indeed true that patients using DPP-4 inhibitors may exhibit less glycemic control overall compared to those on first-line treatment with metformin. However, this reflects a common scenario where both DPP-4 inhibitors and SGLT2 inhibitors are considered for similar patient populations who have not achieved satisfactory glycemic control with metformin alone. This similarity in clinical context helps mitigate the risk of confounding due to differential treatment practices, making DPP-4 inhibitors a relevant comparator for evaluating the effects of SGLT2 inhibitors on cardiovascular outcomes. We have revised our discussion section to provide a clearer understanding and to more accurately reflect this rationale. 

<Revision in the manuscript on page 9-10> 

In our study, we selected DPP4 inhibitors as an active comparator like other prior studies, because they are relatively new and widely used oral antihyperglycemic agents.9 10 Like SGLT2 inhibitors, DPP4 inhibitors are commonly used as second-line therapies after metformin in Korea. 11 Both DPP-4 inhibitors and SGLT2 inhibitors are prescribed under similar clinical circumstances, specifically for patients who do not achieve sufficient glycemic control with metformin alone. This similarity would help reduce the potential for confounding in our evaluation of CV outcomes. In addition, DPP4 inhibitors have shown a neutral effect in most CV outcome trials,12 13 and they have a moderate effect in reducing HbA1c with a low risk of hypoglycemia, similar to SGLT2 inhibitors.14 By choosing DPP-4 inhibitors, we aimed to compare the efficacy and safety of SGLT2 inhibitors against a drug with a similar clinical profile, thereby providing meaningful insights into their relative effects.

4. In the discussion section, the authors used the following statement “Several studies have shown that the use of SGLT2 inhibitors was associated with a reduced risk of incident AF”. I recommend rewriting the sentence avoiding the word “several” as you only cite 2 studies. Reference 24 is a review of individual studies, I suggest citing all of them instead of just citing the review

We appreciate your comments. Reference 24 provides a pooled analysis of CV and renal outcome trials. As suggested by the reviewer, we have replaced the word “several” with “previous” and added more references to support this point. 

Revision in the manuscript on page 11: 

Previous studies have shown that the use of SGLT2 inhibitors was associated with a reduced risk of incident AF. 7,8,24-27 

5. Finally, in the discussion section you make an interesting acknowledgement about the limitation of your study to assess the effect in SGLT-2 inhibitors in non-diabetic patients with AF. I recommend adding a suggestion to further research in this population.

We thank for your invaluable suggestion. We added this in the discussion section as the reviewer suggested. 

Revision in the manuscript on page 12: 

Fourth, we assessed the effects of SGLT inhibitors in diabetic patients with AF. Thus, whether SGLT2 inhibitors improve clinical outcomes in non-diabetic patients with AF is uncertain. Further research is warranted to explore the effects of SGLT2 inhibitors in patients with AF, irrespective of their diabetes status.

Reviewer #2: 

This research presents a retrospective cohort study investigating the efficacy of SGLT2i in patients with type 2 diabetes (T2D) complicated by atrial fibrillation (AF). Leveraging the Korean National Health Insurance Service database, the study assesses the impact of SGLT2i on the composite endpoint of hospitalization for heart failure (HF) or death. The merits of this research lie in: 1) Demonstrating that SGLT2i significantly reduce the risk of HF hospitalization or death among T2D patients with AF, providing novel evidence for the application of SGLT2i in this patient population. 2) Conducting a large-scale retrospective study utilizing the national health insurance database, enhancing the reliability and representativeness of the findings. 3) Implementing propensity score matching for 37 confounding variables, effectively mitigating their influence and ensuring the rigor of the study design. However, I have the following questions regarding this research:

1. The lack of detailed subgroup analyses by different types of AF (e.g., paroxysmal, persistent, or permanent) or by specific SGLT2i limits the generalizability of the results.

We sincerely appreciate all your constructive and detailed comments. 

Due to the nature of the national insurance claims data we analyzed, most cases did not clearly specify the subtype of AF (e.g., paroxysmal, persistent, or permanent). This limitation precluded us from conducting detailed subgroup analyses based on different types of AF. Furthermore, we could not obtain specific data on individual SGLT2 inhibitors; the dataset provided drug information only at the class level to protect personal information. This restriction is in accordance with the national health insurance service's policy, which aims to limit the release of data that could potentially identify individuals and be misused for pharmaceutical marketing purposes. We have included these points in our limitations section of the revised manuscript to enhance clarity.

Revision in the manuscript on page 11: 

First, the national cohort database used in our analysis provides drug information only by class to protect personal information. Consequently, we were unable to assess the effects according to the specific types of each drug used. Also, detailed patient information, AF subtype, and laboratory results were not available, which did not allow us to conduct analyses related to these aspects. Therefore, we could not perform detailed analysis to explore the mechanisms by which SGLT2 inhibitors exert their effects, such as potential associations with blood pressure, lipid profiles, or diuretic effects. 

2. Given that SGLT2i can improve blood pressure, lipid profiles, and exhibit diuretic effects, all of which may contribute to better AF outcomes, could mediation analysis be employed to explore the role of these improvements in mediating the observed effects?

We appreciate your insightful comments regarding the potential influence of SGLT2 inhibitors on blood pressure, lipid metabolism, and diuretic effects contributing to AF outcomes. It is indeed a fascinating area to explore which factors predominantly contribute to better outcome in this population. However, the dataset does not provide detailed information on individual patients' altered indicators, making such mediation analysis challenging. We acknowledged this limitation more specifically in the revised manuscript. 

Revision in the manuscript on page 11: 

First, the national cohort database used in our analysis provides drug information only by the class to protect personal information. Therefore, we were unable to assess the effects according to the specific types of each drug used. Also, detailed patient information, AF subtype, and laboratory results were not available, which did not allow us to conduct analyses related to these aspects. Thus, we could not perform detailed analysis to explore the mechanisms by which SGLT2 inhibitors exert their effects, such as potential associations with blood pressure, lipid profiles, or diuretic effects. 

3. Subgroup analyses based on patient baseline characteristics are crucial to gain a fuller understanding of the efficacy variations of SGLT2i across different patient groups.

We thank for your suggestion. We have added a subgroup analysis based on patient baseline characteristics in Fig 3 of the revised manuscript, as suggested by the reviewer. 

Revision in the manuscript: Figure 3

4. The situation of combination therapy should be carefully elucidated. GLP-1RA use could impact cardiac outcomes. While SGLT2i patients likely have GLP-1RA co-treatment, DPP4i patients should not. Yet, Table 1 shows 8% GLP-1RA co-use in both, raising doubts about a statistical error.

We appreciate your critical comments. Upon re-checking our dataset, we identified a critical error in Table 1, where the orders of medications was incorrectly arranged. Specifically, the prescription rate of GLP-1RA was 0.3% in both the SGLT2 inhibitors and DPP4 inhibitors group. 

Thes low co-use rate of SGLT2i and GLP1RA can be attributed to specific insurance guidelines in Korea. In our country, most medical costs are covered by the national health insurance system, which sets the reimbursement criteria for medication prescriptions. Currently, the combination use of SGLT2i and GLP-1RA is not reimbursed in Korea, resulting in a financial burden for patients who would need to pay out-of-pocket for these medications. This financial constraint likely contributes to the low rate of combination therapy.

Similarly, the combination of DPP4 inhibitors and GLP-1RA is not reimbursed and is actually contraindicated. However, in some cases, both medications might be prescribed together. For instance, a patient already on DPP4 inhibitor for diabetes management may be prescribed a GLP-1RA either for weight loss or to achieve better glycemic control, possibly by a different healthcare provider or due to an oversight in clinical judgment.

We have corrected Table 1 accordingly in the revised manuscript. Once again, we deeply appreciate the reviewer’s all the detailed and constructive comments and suggestions. 

Revision in Table 1

---

## [Decision Letter · Decision Letter 1]

15 Oct 2024

PONE-D-24-09659R1The effect of SGLT2 inhibitor in patients with type 2 diabetes and atrial fibrillationPLOS ONE

Dear Dr. Shin,

Thank you for submitting your manuscript to PLOS ONE. After careful consideration, we feel that it has merit but does not fully meet PLOS ONE’s publication criteria as it currently stands. Therefore, we invite you to submit a revised version of the manuscript that addresses the points raised during the review process.

We look forward to receiving your revised manuscript.

Kind regards,

Xinlin Zhang

Academic Editor

PLOS ONE

Journal Requirements:

Reviewers' comments:

Reviewer's Responses to Questions

**Comments to the Author**

1. If the authors have adequately addressed your comments raised in a previous round of review and you feel that this manuscript is now acceptable for publication, you may indicate that here to bypass the “Comments to the Author” section, enter your conflict of interest statement in the “Confidential to Editor” section, and submit your "Accept" recommendation.

Reviewer #1: (No Response)

Reviewer #2: All comments have been addressed

2. Is the manuscript technically sound, and do the data support the conclusions?

Reviewer #1: Yes

Reviewer #2: Yes

3. Has the statistical analysis been performed appropriately and rigorously? 

Reviewer #1: Yes

Reviewer #2: Yes

4. Have the authors made all data underlying the findings in their manuscript fully available?

Reviewer #1: Yes

Reviewer #2: Yes

5. Is the manuscript presented in an intelligible fashion and written in standard English?

Reviewer #1: Yes

Reviewer #2: Yes

6. Review Comments to the Author

Reviewer #1: 1. The authors have made significant improvements to the presentation of subgroup results. However, the revised statement indicates that the subgroup analysis demonstrates a causal interaction. This finding should also be addressed in the methods section, particularly regarding the interpretation of the p-value for interaction. To further elucidate this observation, I recommend the following article: Brankovic M, Kardys I, Steyerberg EW, et al. "Understanding Interaction (Subgroup) Analysis in Clinical Trials." European Journal of Clinical Investigation. 2019; 49:e13145. https://doi.org/10.1111/eci.13145.

Reviewer #2: The authors have thoroughly addressed my previous comments and suggestions. The changes made have significantly improved the clarity, rigor, and overall quality of the manuscript. I have no further concerns and believe that the manuscript is now suitable for publication in this journal.

7. PLOS authors have the option to publish the peer review history of their article (what does this mean?). If published, this will include your full peer review and any attached files.

Reviewer #1: **Yes: **Marcelo Reategui-Diaz

Reviewer #2: **Yes: **Xiaowen Zhang

---

## [Author Response · Author response to Decision Letter 1]

3 Nov 2024

Reviewer #1: 1. The authors have made significant improvements to the presentation of subgroup results. However, the revised statement indicates that the subgroup analysis demonstrates a causal interaction. This finding should also be addressed in the methods section, particularly regarding the interpretation of the p-value for interaction. To further elucidate this observation, I recommend the following article: Brankovic M, Kardys I, Steyerberg EW, et al. "Understanding Interaction (Subgroup) Analysis in Clinical Trials." European Journal of Clinical Investigation. 2019; 49:e13145. https://doi.org/10.1111/eci.13145.

We would like to thank you for your valuable comments and suggestions, which have greatly contributed to the improvement of this manuscript. We added/modified the text in the method section and result section as per your suggestion.

Method - Statistical analysis

After propensity score matching, we performed survival analyses to estimate the effect of SGLT2 inhibitors on clinical outcomes. The Kaplan–Meier estimates were performed. The difference between the survival curves was assessed by the log-rank test. A marginal Cox proportional hazards regression models for a cluster in a matched pair were performed. We assessed the consistency of the drug effect on the primary outcome in subgroups. Subgroup analyses were performed to assess whether the effect of SGLT2 inhibitor on clinical outcomes varied across different baseline characteristics. We computed interaction p-values to detect effect modification. An interaction p-value < 0.05 would indicate a statistically significant interaction effect between the treatment and the subgroup variable, suggesting the presence of effect modification, but it is important to note that non-significant interaction p-values do not rule out the possibility of clinically relevant differences in subgroups.1 In addition, to evaluate whether the benefit of the SGLT2 inhibitor varied with the follow-up period after the time of initiation, analyses were performed according to the time after initiation of the drug (30, 90, 180 days, 1, and 3 years after the index date). Two-sided p values < 0.05 were considered statistically significant. All analyses were performed with SAS (ver. 9.4; SAS Institute, Cary, NC, USA) and R software (ver. 4.0.3; R Development Core Team, Vienna, Austria).

Result

The effect of the SGLT2 inhibitor on the primary outcome was consistent across all subgroups, indicating that SGLT2 inhibitors were associated with a lower risk of the composite of HF hospitalization or death compared to DPP4 inhibitors, regardless of age, gender, comorbidities, prior history of atherosclerotic CV disease or HF, and the use of beta blocker or renin-angiotension system blocker (all p for interaction>0.05, Fig 3). This suggests that the impact of SGLT2 inhibitor relative to the DPP-4 inhibitor was consistent across the baseline characteristics evaluated. However, this should not be interpreted as definitive evidence of uniform effects across all subgroups. Further studies with more power or focusing on specific subgroups may provide additional insights. Patients with SGLT2 inhibitor demonstrated a significantly lower risk of primary outcome from 90 days after initiating the drug to the third year, as compared with those with DPP4 inhibitor (P<0.05, Table 3).

Reference

1 Brankovic M, Kardys I, Steyerberg EW, et al. Understanding of interaction (subgroup) analysis in clinical trials. European journal of clinical investigation 2019;49:e13145.

---

## [Editor Report · Decision Letter 2]

12 Nov 2024

The effect of SGLT2 inhibitor in patients with type 2 diabetes and atrial fibrillation

PONE-D-24-09659R2

Dear Dr. Shin,

We’re pleased to inform you that your manuscript has been judged scientifically suitable for publication and will be formally accepted for publication once it meets all outstanding technical requirements.

Kind regards,

Xinlin Zhang

Academic Editor

PLOS ONE

---

## [Editor Report · Acceptance letter]

15 Nov 2024

PONE-D-24-09659R2 

PLOS ONE

Dear Dr. Shin, 

I'm pleased to inform you that your manuscript has been deemed suitable for publication in PLOS ONE. Congratulations! Your manuscript is now being handed over to our production team.

Kind regards, 

on behalf of

Professor Xinlin Zhang 

Academic Editor

PLOS ONE